*Experimental Results* (2021), 2, e6, 1–7

CAMBRIDGE
UNIVERSITY PRESS

**LIFE SCIENCE AND BIOMEDICINE**

**NOVEL-RESULT**

# Rapid shortening of survival duration in early fatal cases of COVID-19, Wuhan, China

Yuzhou Zhang, PhD[1] (ORCID), Hilary Bambrick, PhD[1], Shilu Tong, PhD[1,2], Stephen B Lambert, MBBS, PhD[3] (ORCID) and Wenbiao Hu, PhD[1,*] (ORCID)

[1]School of Public Health and Social Work, Institute of Health and Biomedical Innovation, Queensland University of Technology, Brisbane 4059, Queensland, Australia, [2]Shanghai Children's Medical Center, Shanghai Jiao Tong University School of Medicine, Shanghai 200127, China, and [3]National Centre for Epidemiology and Population Health, The Australian National University, Canberra 0200, Australian Capital Territory, Australia
*Corresponding author: Email: w2.hu@qut.edu.au

(Received 09 December 2020; Revised 31 December 2020; Accepted 31 December 2020)

### Abstract

Severe COVID-19 cases place immediate pressure on hospital resources. To assess this, we analysed survival duration in the first 39 fatal cases in Wuhan, China. Time from onset and hospitalization to death declined rapidly, from ~40 to 7 days, and ~25 to 4 days, respectively, in the outbreak's first month.

**Keywords:** COVID-19; survival duration; fatal cases; Wuhan

## 1. Introduction

In late December 2019 it was reported the capital of Hubei province in China, Wuhan city, was experiencing an outbreak of pneumonia of unknown cause (ProMED-mail, 2019). A novel coronavirus, now named SARS-CoV-2, was identified as the causative agent on 07 January 2020.

Since then spread to more than 200 other countries has occurred, resulting in more than 95 million cases and over 2 million deaths (Johns Hopkins University, 2020). Where uncontrolled transmission has occurred, healthcare systems are quickly overwhelmed, resulting in a restricted capacity to care for the sickest patients. The response to viral pandemics requires rationing of scarce medical capital, including equipment, expertise, and interventions (Emanuel et al., 2020).

Disease due to SARS-CoV-2 is severe in a relatively high proportion of infected adults. In an early report from Wuhan city involving 710 cases, 52 (7.3%) patients required ventilation and intensive care unit (ICU) management, and 32 (4.5%) had died by day 28. In countries with large outbreaks, including Italy and Spain, capacity to manage the sickest patients, by ventilation in an ICU bed, was exhausted quickly (Macintyre & Heslop, 2020). The Italian College of Anesthesia, Analgesia, Resuscitation, and Intensive Care published guidelines recognising not everyone could be optimally managed, and recommending triage following "the most widely shared criteria regarding distributive justice and the appropriate allocation of limited health resources" (Mounk, 2020).

As available resources are consumed and exhausted, time to death in fatal cases from readily available timepoints could be monitored as a simple measure of system stress. The most obvious of these include, where available, time from illness onset to death and time from hospitalisation to death.

We sought to identify available information from Wuhan city to understand how the pressure of early cases had an impact on these outcomes early in the outbreak.

## 2. Methods

Core data on the first 39 fatal cases of COVID-19 in Wuhan city were publicly available (The National Health and Safety Commission of the People's Republic of China, n.d.a; n.d.b; n.d.c; n.d.d). For each case we retrieved age in years, gender, and dates of illness onset, hospitalization, and death.

To better understand the impact on patient management of overwhelmed systems, we used a generalized linear regression model with Poisson link to explore changes in the time between symptom onset and death, and hospitalization and death, for first 39 fatal cases of COVID-19 in Wuhan (up to 25 January 2020).

We fitted the model as follows:

$$\log(u_t) = \beta_0 + \beta_1 x_1 + \beta_2 x_2 + \beta_3 x_3 + e_t.$$

where, $u_t$ is the time (days) between the dates of symptom onset/hospitalization and death; $\beta_0$ is the intercept for the model;

$x_1$ is the time (days) between the dates of symptom onset/hospitalization and these dates in the first reported case;

$x_2$ is gender; $x_3$ is age (years);

$\beta_1, \beta_2, \beta_3$ are the corresponding regression coefficients for these independents; and

$e_t$ is the error term.

For three cases, only approximate dates for onset of illness were available. We undertook a sensitivity analysis by repeating our work excluding these cases.

We used values generated in the model to calculate the average rate of day-by-day change in survival duration during the outbreak.

Ethics committee approval was not sought as we analysed routinely collected, publicly available de-identified data.

## 3. Results

In the earliest cases resulting in death due to COVID-19 in Wuhan city, we found later cases had a shortened survival period compared to earlier fatal cases. Days from symptom onset to death and days from hospitalization to death both declined significantly and rapidly early in the outbreak (Figure 1).

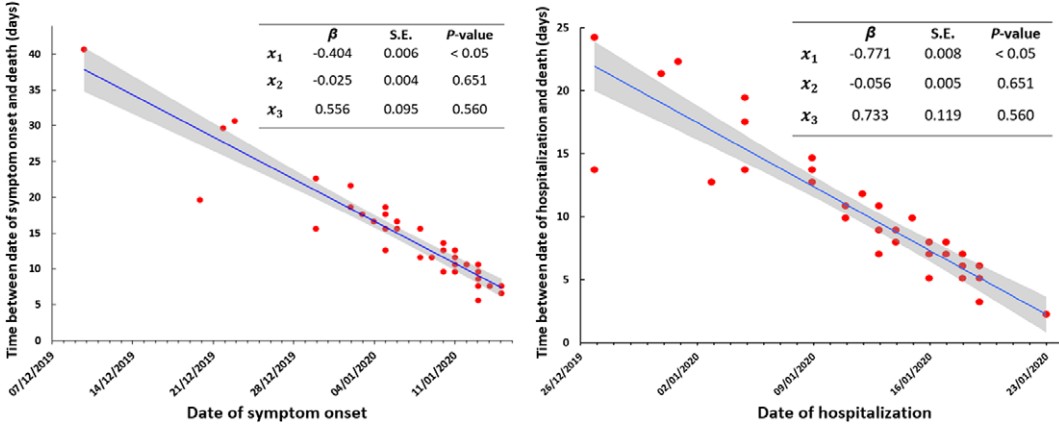

**Figure 1.** Time between date of symptom onset and death (left panel) and hospitalization and death (right panel) by date for the first 39 fatal cases of COVID-19, Wuhan city, China.x1 is the time (days) between the dates of symptom onset/ hospitalization and death in the first reported case; x2 is gender; x3 is age (years); β is the corresponding regression coefficient for the independent values; S.E. is the standard error of the regression coefficient; shading represents the 95% confidence interval around the regression line.

Using these data, the median (mean) time from symptom onset to death was 13 days (15.1 days), and for hospitalization to death, 8 days (9.2 days). The time from symptom onset to death fell from ~40 days to approximately ~7 days. The time from hospitalization to death were ~ 25 days to ~4 days.

The average rate of decline from symptom onset to death in fatal cases was 0.40 days per day (95% confidence interval (CI): 0.28–0.52) in little over one month, with the equivalent reduction from hospitalization to death being 0.77 days per day (95% CI: 0.63–0.91).

Consistent results were found when we excluded three fatal cases with only approximate onset dates (data not shown).

## 4. Discussion

We found a rapid shortening of survival duration in fatal cases early in the Wuhan COVID-19 outbreak. This likely reflects the increased stress placed on local health resources prior to the implementation of community-wide control measures, preventing optimal support for many of the sickest patients. Over the first month of the outbreak, our models suggest the time from symptom onset to death fell from 40 to 7 days, and from hospitalization to death from 25 to 4 days. A similar finding was found in the UK, which indicated that the unadjusted survival at 30 days was lowest for people admitted in both high dependency unit and ICU in late March, the early stage of the epidemic in the country (Dennis et al., 2020).

SARS-CoV-2 is highly transmissible with current $R_0$ estimates between 2 and 3 (Wu et al., 2020), but this value may be as high as 5.7 is settings without prior cases or implemented control measures (Sanche et al., 2020). Uncontrolled transmission results in a rapid increase in case numbers, outpatient visits, and hospital and ICU admissions. To prevent the rapid overwhelming of finite health resources, controlling SARS-CoV-2 transmission early in local outbreaks is critical. Pre-emptive low-cost social distancing and enhanced hygiene activities and, where appropriate, other community-based controls—quarantine, city lockdowns with school and work closures, cancelling mass gatherings—should be considered. Deferring, and thus reducing, the early peak in cases will reduce the likelihood of health systems being overwhelmed immediately, allowing better management of those patients that do become ill, with likely improved clinical outcomes.

Our analysis is limited to available data from small numbers of fatal Wuhan cases early in the outbreak. System pressures may affect the quality of onset date data but should have no impact on dates of hospitalisation and death. If, over time, infected patients presented increasingly later in their clinical course, this could artifactually result in a progressively shorter time to death. However, as seen in first 100 cases in Singapore, time to hospitalisation fell quickly in the early stages of the outbreak (Ng et al., 2020).

The data on time to death were positively skewed and changed quickly over time. At the start of an epidemic, these summarised data hold important details for other countries and regions without cases. They clearly demonstrate not only that a relatively high proportion of cases are severe, requiring intensive care, but also provide some guidance on how quickly existing systems for such care are overwhelmed. Where made easily available, key dates from early severe cases, including data of symptom onset, hospitalisation, ICU admission, and death, should be made publicly available. Of even more benefit would be if they were available alongside details of rationing of ICU and other resources.

Using publicly available data, we were able to demonstrate a dramatic shortening of time from illness onset and hospitalization to death in early COVID-19 cases in Wuhan city, China. Our findings highlight that in pandemics, systems of care are rapidly overwhelmed. Time from symptom onset and hospitalisation to death could be used as key pandemic metrics to monitor deterioration in the healthcare system's ability to manage the sickest of patients, and to also track subsequent return to baseline.

**Acknowledgements.** The authors thank the National Health Commission of the People's Republic of China for making the data used in this analysis publicly available.

We gratefully acknowledge the efforts of healthcare workers in Wuhan, and globally, caring for patients infected with SARS-CoV-2.

**Author Contributions.** Zhang and Hu had full access to all the data in the study and take responsibility for the integrity of the data and the accuracy of the data analysis.

Study concept and design: Hu.

Analysis and interpretation of data: All authors.

Drafting of the manuscript: Zhang.

Critical revision of the manuscript for important intellectual content: Bambrick, Tong, Lambert, Hu.

Statistical analysis: Zhang and Hu.

Study supervision: Hu.

**Funding Information.** This research received no specific grant from any funding agency, commercial or not-for-profit sectors.

**Data Availability Statements.** All COVID-19 fatal cases data used in the study were declare in the methods section.

**Conflict of Interest.** The authors report no potential conflicts of interest.

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

---

# Peer Reviews

**Reviewing editor:** Dr. Michael Nevels

University of St Andrews, Biomolecular Sciences Building, Fife, United Kingdom of Great Britain and Northern Ireland, KY16 9ST

This article has been accepted because it is deemed to be scientifically sound, has the correct controls, has appropriate methodology and is statistically valid, and has been sent for additional statistical evaluation and met required revisions.

doi:10.1017/exp.2020.73.pr1

## Review 1: Rapid shortening of survival duration in early fatal cases of COVID-19, Wuhan, China

**Reviewer:** Dr. Sher Bahadur Pun 

Sukraraj Tropical & Infectious Disease Hospital, Nepal

Date of review: 25 December 2020

**Conflict of interest statement.** Reviewer declares none

*Comments to the Author:* 1. Is ut and x1 denote same meaning ? or x1 denote a single first reported case only?

2. Did the authors overlap the cases who first showed symptoms then recount same cases after hospitalizations? Because the authors drew two separate figures, one is from symptom onset to death and another is hospitalization to death

3. Do these results compatible with findings from other countries?

4. What is the significant of this study? It is obvious that any new communicable disease (during outbreak) are usually overwhelmed hospitals with patients and even deaths.

## Score Card

### Presentation

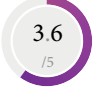

| | |
|---|---|
| Is the article written in clear and proper English? (30%) | 4/5 |
| Is the data presented in the most useful manner? (40%) | 3/5 |
| Does the paper cite relevant and related articles appropriately? (30%) | 4/5 |

3.6 /5

### Context

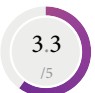

| | |
|---|---|
| Does the title suitably represent the article? (25%) | 3/5 |
| Does the abstract correctly embody the content of the article? (25%) | 4/5 |

3.3 /5

Does the introduction give appropriate context? (25%)     3/5

Is the objective of the experiment clearly defined? (25%)     3/5

## Analysis

3.0
/5

Does the discussion adequately interpret the results presented? (40%)     3/5

Is the conclusion consistent with the results and discussion? (40%)     3/5

Are the limitations of the experiment as well as the contributions
of the experiment clearly outlined? (20%)     3/5

doi:10.1017/exp.2020.73.pr2

# Review 2: Rapid shortening of survival duration in early fatal cases of COVID-19, Wuhan, China

**Reviewer:** Dr. Selda Tekiner

Date of review: 27 December 2020

**Conflict of interest statement.** Reviwer declares none

*Comments to the Author:* Thanks to the authors

---

## Score Card

### Presentation

**5.0** /5

| | |
|---|---|
| Is the article written in clear and proper English? (30%) | 5/5 |
| Is the data presented in the most useful manner? (40%) | 5/5 |
| Does the paper cite relevant and related articles appropriately? (30%) | 5/5 |

### Context

**5.0** /5

| | |
|---|---|
| Does the title suitably represent the article? (25%) | 5/5 |
| Does the abstract correctly embody the content of the article? (25%) | 5/5 |
| Does the introduction give appropriate context? (25%) | 5/5 |
| Is the objective of the experiment clearly defined? (25%) | 5/5 |

### Analysis

**5.0** /5

| | |
|---|---|
| Does the discussion adequately interpret the results presented? (40%) | 5/5 |
| Is the conclusion consistent with the results and discussion? (40%) | 5/5 |
| Are the limitations of the experiment as well as the contributions of the experiment clearly outlined? (20%) | 5/5 |