## [Reviewer Report · Review 1: Rapid shortening of survival duration in early fatal cases of COVID-19, Wuhan, China]

*Comments to the Author:* 1. Is ut and x1 denote same meaning ? or x1 denote a single first reported case only?

2. Did the authors overlap the cases who first showed symptoms then recount same cases after hospitalizations? Because the authors drew two separate figures, one is from symptom onset to death and another is hospitalization to death

3. Do these results compatible with findings from other countries?

4. What is the significant of this study? It is obvious that any new communicable disease (during outbreak) are usually overwhelmed hospitals with patients and even deaths.

## Score Card

### Presentation

3.6/5

Is the article written in clear and proper English?30%4/5

Is the data presented in the most useful manner?40%3/5

Does the paper cite relevant and related articles appropriately?30%4/5

### Context

3.3/5

Does the title suitably represent the article?25%3/5

Does the abstract correctly embody the content of the article?25%4/5

Does the introduction give appropriate context?25%3/5

Is the objective of the experiment clearly defined?25%3/5

### Analysis

3.0/5

Does the discussion adequately interpret the results presented?40%3/5

Is the conclusion consistent with the results and discussion?40%3/5

Are the limitations of the experiment as well as the contributions of the experiment clearly outlined?20%3/5

---

## [Reviewer Report · Review 2: Rapid shortening of survival duration in early fatal cases of COVID-19, Wuhan, China]

*Comments to the Author:* Thanks to the authors

## Score Card

### Presentation

5.0/5

Is the article written in clear and proper English?30%5/5

Is the data presented in the most useful manner?40%5/5

Does the paper cite relevant and related articles appropriately?30%5/5

### Context

5.0/5

Does the title suitably represent the article?25%5/5

Does the abstract correctly embody the content of the article?25%5/5

Does the introduction give appropriate context?25%5/5

Is the objective of the experiment clearly defined?25%5/5

### Analysis

5.0/5

Does the discussion adequately interpret the results presented?40%5/5

Is the conclusion consistent with the results and discussion?40%5/5

Are the limitations of the experiment as well as the contributions of the experiment clearly outlined?20%5/5